# Assessment of Psychosocial Correlates and Associated Factors of Colorectal Cancer Screening among Southwestern Saudi Population: A Cross-Sectional Study

**DOI:** 10.3390/healthcare11202791

**Published:** 2023-10-21

**Authors:** Anfal Mohammed Alenezi, Mahadi Mane Hussien Alshariyah, Maryam Nazal Alanazi, Doaa Mazen Abdel-Salam, Ahmad Homoud Al-Hazmi, Ashokkumar Thirunavukkarasu, Ahmed M. Alhuwaydi, Rahaf Hamdan Alsabilah, Rehab A. Mohamed

**Affiliations:** 1Department of Surgery, College of Medicine, Jouf University, Sakaka 72388, Saudi Arabia; 2Department of Public Health, Najran Health Affairs, Najran 66232, Saudi Arabia; mahadi1403@gmail.com; 3Department of Psychological Counseling, College of Sciences and Arts, Qurrayat Campus, Jouf University, Qurrayat 77425, Saudi Arabia; mnalenazi@ju.edu.sa; 4Department of Public Health and Community Medicine, Faculty of Medicine, Assiut University, Assiut 71515, Egypt; doaa.mazen@aun.edu.eg; 5Department of Family and Community Medicine, College of Medicine, Jouf University, Sakaka 72388, Saudi Arabia; ahhazmi@ju.edu.sa (A.H.A.-H.); ashokkumar@ju.edu.sa (A.T.); 6Department of Internal Medicine, Division of Psychiatry, College of Medicine, Jouf University, Sakaka 72388, Saudi Arabia; amalhuwaydi@ju.edu.sa; 7College of Medicine, Jouf University, Sakaka 72388, Saudi Arabia; raahaf.alsabila@gmail.com; 8Department of Family Medicine, Faculty of Medicine, Suez Canal University, Ismailia 41522, Egypt; rehabali@med.suez.edu.eg

**Keywords:** colorectal cancer, screening, cancer worries, social influence, Saudi Arabia

## Abstract

Psychosocial correlates are one of the crucial determinants for the uptake of colorectal cancer (CRC) screening by the pre-eligible population. The present study aimed to identify the psychosocial correlates of colorectal cancer screening and determine their associated factors among the Saudi population in the Najran region, Saudi Arabia. Using a validated questionnaire, we assessed five constructs of psychosocial correlates of CRC screening among 790 participants aged 45 years and above. The five constructs were salience and coherence, cancer worries, perceived susceptibility, response efficacy, and social influence. Of the studied population, less than 50% agreed with most of the five constructs’ statements, and 27.5% preferred to follow their family members’ advice. Significantly higher mean scores (±SD) were identified for the male gender (7.38 ± 2.15, *p* = 0.027) and participants working in government sectors (7.60 ± 2.03, *p* = 0.027) in the cancer worries construct, while the mean (±SD) scores of perceived susceptibility were significantly higher among married participants (14.38 ± 4.10, *p* = 0.023) and smokers (14.95 ± 3.92, *p* = 0.041). Our survey results could help policymakers to implement focused health education programs for the pre-eligible population on the importance of the uptake of CRC screening. Furthermore, it is recommended to carry out exploratory mixed-method surveys in other regions of Saudi Arabia to understand the region’s specific psychosocial correlates towards CRC screening.

## 1. Introduction

The world health organization (WHO) states that cancer is the second commonest cause of mortality and contributes to 9.6 million deaths per year globally [1]. Colorectal cancer (CRC) is the second commonest occurring cancer in women and the third most frequently developing cancer in men. CRC has become a significant burden on the healthcare systems of many countries due to its increasing incidence, attributed to the change in lifestyles and nutrition habits [1,2]. Recent epidemiological surveys in the Kingdom of Saudi Arabia (KSA) reported that CRC is the most commonly occurring cancer in men and the third most commonly occurring cancer in women [3,4].

With this increasing global burden of CRC, prevention will play a vital role in handling this critical public health challenge of the 21st century. Several national and international organizations have suggested that CRC screening is a powerful method by which the early stage of CRC (precancerous polyps and growths) can be detected [2,5,6]. Community-based colorectal screening services are feasible only in economically developing countries, but potential focus should also be given to those parts of the world with an aging population and a growing Westernized lifestyle [7]. This disease is highly curable in its early stage because of the slow development of CRC, and screening will minimize the incidence and mortality of CRC. For men and women at an average risk, regular screening for CRC is recommended from the age of 50 years [8]. CRC screening remains the most relevant and cost-effective method to minimize the mortality and morbidity of this disease, while both risk factor reductions and enhanced therapies contribute less to it [9]. A study conducted by Ghoncheh M et al. stated that the occurrence of CRC and associated deaths is positively related to the human development index (HDI) [10]. According to the United Nations Development Program, the KSA comes under a very high HDI category. This indicates the importance of CRC in the KSA for preventing CRC incidence and decreasing mortality through early diagnosis and necessary intervention [11]. Recent research in Saudi Arabia recorded that 34.56% of diagnosed CRC patients had metastatic involvement, 76% had late-stage CRC, and the majority of late-stage cases were female. The results showed the diagnostic differences in the CRC process in the Saudi population [12].

Broadly speaking, the screening approaches for CRC include the targeting of non-invasive fecal occult blood samples and invasive endoscopy-based diagnostic studies (flexible sigmoidoscopy and complete colonoscopy). Colonoscopy, on the other hand, is usually considered the gold standard screening method for detecting CRC and precancerous neoplasms and has proven effective in reducing CRC incidence and mortality [13,14]. Along with evidence of an increased incidence of CRC among the young age group (<50) and the young Saudi population, this condition may pressure the national healthcare system over the next few decades [15]. In the KSA, a CRC early detection program has been implemented by the ministry of health to reduce mortality due to CRC through early diagnosis and timely referral for necessary management. All the healthcare services, including the CRC screening program, are given free of cost to Saudi nationals and expatriates affiliated with government sectors. They are pre-eligible to uptake the CRC screening service provided at designated health centers.

Theories of health behavior often indicate that attitudes related to health control can be major determinants of behaviors that improve health [16]. In the literature on health promotion and disease prevention, including cancer-preventive behaviors, intention has been recognized as the most important predictor of the involvement of individuals in a given health behavior [17]. Using an instrument adapted from the preventive health paradigm, Salimzadeh et al. concluded that most respondents reported poor attitudes toward CRC screening [18]. Shokar et al., inspired by the Health Belief Model, found that self-efficacy, perceived advantages, and fatalism may be targeted to enhance the effectiveness of CRC screening initiatives [19]. Significantly lower degrees of response efficacy, higher perceived barriers, and poorer levels of self-efficacy were all social cognitive determinants of non-adherence to CRC screening, as mentioned by a study detecting behavioral and demographic predictors of adherence to CRC screening tests [20]. The integrated screening action model (I-SAM) presents a comprehensive framework to assist in comprehending screening behavior and finding intervention targets. It describes six factors: automatic motivation, reflective motivation, psychological capability, physical capability, social opportunity, and physical opportunity. These factors can impede or enhance cancer screening throughout individual and contextual contexts [21]. Irrespective of their screening background, a different study found that patients who wished to undergo colorectal screening encountered the same kinds of barriers; in addition, they agreed on the most significant obstacles [22]. After an extensive study of the literature, we found that there is deficient information about the psychosocial correlates of colon cancer screening among the Saudi population in the Najran Region, Saudi Arabia. The present study aimed to assess the psychosocial correlates and associated factors of colorectal cancer screening among the Saudi population in the Najran region, Saudi Arabia.

## 2. Materials and Methods

### 2.1. Study Design and Setting

The present cross-sectional study was conducted from April 2022 to November 2022 in the Najran Region of the KSA, situated in the Southwestern part of the KSA.

### 2.2. Sample Size Estimation and Sampling Method

The sample size was calculated using Cochran’s sample size estimation formula (*n* = z^2^ pq/e^2^) [23]. The authors determined the following criteria after reviewing the available literature to calculate the lowest sample size: expected frequency (p) of 50%, confidence level of 95% (z = 1.96), design effect of 2, and margin of error (e) of 5%. Applying the above-mentioned values and criteria, the calculated sample size was 790 participants (rounded). The required number of participants for the present study were recruited from the 10 primary health centers in the Najran region, KSA. Selection of the 10 primary health centers was carried out via a simple random sampling technique among 45 centers in the Najran region, KSA. The number of study participants selected in each center was proportional to the number of clients of health services served by this center until it reached the required sample size.

### 2.3. Inclusion and Exclusion Criteria

The present study included male and female participants aged 45 years and above from the Najran region. Participants who were not willing to participate and those who had a history of colorectal cancer and polyps were excluded from the survey. Also, participants who had undergone CRC screening earlier were excluded from the current survey. Participants in the present study were approached during their visits to the PHC for seeking any health services offered at the PHC, such as for the vaccination of their children, follow-ups for chronic diseases, or other health services.

### 2.4. Data Collection Method

Firstly, the invited participants were given a brief orientation about different CRC screening methods, the type of screening method recommended by the ministry of health, KSA, for high-risk groups, and the current survey’s purpose. Informed consent was obtained from the study participants who were willing to participate. They were asked to fill in the electronic data collection sheet (Google form) using the data collectors’ personal digital devices.

A pretested and validated survey questionnaire in Arabic was given to all the study participants. This survey tool has been used in primary care settings and the use of the scale is supported among different populations and cultural backgrounds [24,25,26,27]. The research team obtained the required copyrights from the American Association for Cancer Research to translate the survey into Arabic. We followed the standard protocols for translating the adapted questionnaire [28,29,30]. Three independent translators executed the initial forward translation from English to Arabic: two from the medical field (community medicine and surgery department) and one from the non-medical field (native Arabic speaker). After forward translation, the translators had a focus group discussion about making a single version of the Arabic questionnaire. Two other translators performed the back-translation from Arabic to English to ensure that there were no misunderstanding and unclear words. We performed a pilot test among 30 participants aged 50 years with the translated questionnaire. All pilot test participants stated that the questionnaire was simple, suitable to local settings, and easy to understand.

The questionnaire is divided into two parts:First part: This part included socio-demographic characteristics such as age, sex, educational status, working status, marital status, family history of colorectal cancer, and health insurance.Second part: This part consisted of a scale intended to evaluate psychosocial determinants associated with CRC screening. The questionnaire measured five constructs: salience and coherence, cancer worries, perceived susceptibility, response efficacy, and social influence. Salience and coherence is defined as the feeling that carrying out a health-related practice is harmonious with one’s beliefs (4 items, α = 0.91); cancer worries are stated as concerns about unfavorable consequences of participating in CRC screening (2 items, Cronbach’s alpha [α] = 0.74); perceived susceptibility is defined as subjective individual risk for developing CRC after the screening test (4 items, α = 0.79); response efficacy is defined as the belief that applying a health practice will be beneficial in decreasing the possibility of developing the disease (2 items, α = 0.73); and social influence is defined as anticipated beliefs about and willingness to adhere to society’s attitudes toward CRC screening (4 items, α = 0.81). The pre-eligible participants who responded to the sixteen CRC screening psychosocial correlates were evaluated on a 5-point Likert scale. We gave scores ranging from “strongly disagree” (1) to “strongly agree” (5) for each item.

### 2.5. Statistical Analysis

The Statistical Package for Social Sciences (SPSS version-21, Armonk, NY, USA: IBM Corp) was used to export data from the spreadsheet, coding, and analysis. The quantitative data of the present study are expressed as mean ± standard deviation, while qualitative data are expressed as frequencies and proportions. Initially, the data were tested for normal distribution using the Shapiro–Wilk test for normality assumption. The present study consisted of three types of independent variables: dichotomous (variables with two independent groups), categorical (variables with more than two independent groups), and quantitative (continuous). The mean scores of each construct were compared with the socio-demographic and health-related characteristics using the independent *t*-test for dichotomous variables, one-way analysis of variance (ANOVA) for categorical variables, and Pearson’s correlation test for the quantitative variable. Finally, we applied multiple regression analysis for each psychosocial construct after adjusting the independent variables. The present study set the alpha (α) value as 0.05 to identify significant associations between dependent and independent variables. We used a two-tailed statistical test in the present study.

### 2.6. Ethical Consideration

After obtaining ethical clearance from the regional ethics committee (IRB, Najran region, Saudi Arabia—Log no: 2022-07 E) and other required approvals from concerned authorities, the survey team initiated the data collection process. Survey participation was voluntary, and none of the items from the data collection form inquired about participants’ identification details. Hence, we maintained the anonymity and confidentiality of the data.

## 3. Results

Table 1 depicts the socio-demographic and health-related characteristics of the studied population. Of the 790 studied participants, the majority (58.7%) were males with 59.06 ± 6.46 (mean ± SD). Nearly half (46.7%) of them were married, 60.9% had a formal education (high school level and above), 28.4% of them worked in the government sector, and the majority (63.7%) of them resided in an urban area. Regarding health-related characteristics, 72.3% of the participants were non-smokers, and nearly one-third (32.3%) of them were diagnosed with one or more chronic diseases.

Table 2 shows that 38.4% of the participants strongly agreed that CRC makes sense to them, 29.1% of respondents stated that CRC could help them be healthy, and 25.2% of them had concerns that CRC may show that they have colorectal cancer or polyps. Regarding the social influence construct, 27.5% of respondents strongly agreed with the statement “I want to do what members of my immediate family think I should do about colorectal cancer screening”.

Table 3 shows the relationships of the socio-demographic and health-related characteristics with the salience and coherence construct and cancer worries construct. Of the 790 respondents aged 50 years and above, the mean (±SD) scores were significantly higher among the male gender (15.23 ± 4.14, *p* = 0.001), participants working in government sectors (15.62 ± 3.82, *p* = 0.03), and those with a monthly income of more than SAR 7000 (15.72 ± 3.78, *p* = 0.043). Furthermore, Pearson’s correlation test revealed a positive correlation between age and the mean score of the salience and coherence construct (r = 0.631, *p* = 0.01). Regarding the cancer worries construct, significantly higher mean scores (±SD) were identified for the male gender (7.38 ± 2.15, *p* = 0.027) and the participants working in government sectors (7.60 ± 2.03, *p* = 0.027).

The relationships of the socio-demographic and health-related characteristics with the perceived susceptibility and response efficacy constructs are presented in Table 4. Of the 790 sample participants, the mean (±SD) scores were significantly higher among male gender (14.63 ± 4.17, *p* = 0.012), married participants (14.38 ± 4.10, *p* = 0.023), those who live in urban areas (14.65 ± 4.01, *p* = 0.003), and smokers (14.95 ± 3.92, *p* = 0.041). Regarding the response efficacy construct, the mean (±SD) scores were significantly higher among the participants residing in urban areas (7.67 ± 2.02, *p* = 0.001) and associated with other chronic diseases (7.66 ± 2.01, *p* = 0.039).

Regarding the social influence construct, the mean (±SD) scores were significantly higher among male gender (15.34 ± 4.01, *p* < 0.001), married participants (15.25 ± 3.91, *p* = 0.043), government sector employees (15.64 ± 3.72, *p* = 0.008), and those residing in urban areas (15.25 ± 3.89, *p* = 0.001). Furthermore, a positive correlation was found between the age of the participants and the mean score of the social influence construct (r = 0.564, *p* = 0.021) (Table 5).

The regression analysis revealed that the salience and coherence construct is significantly associated with age (*p* = 0.014), gender (0.017), and occupation (*p* = 0.002); the cancer worries construct is significantly associated with gender (*p* = 0.001) and occupation (*p* = 0.030); and perceived susceptibility is significantly associated with gender (*p* = 0.022), education level (*p* = 0.049), occupation (*p* = 0.008), residence (*p* = 0.024), and smoking status (*p* = 0.033). The response efficacy construct is significantly associated with gender (*p* = 0.047) and residence status (*p* = 0.001), and the social influence construct is significantly associated with gender (*p* = 0.005) and residence status (*p* = 0.002) (Table 6).

## 4. Discussion

There is little research investigating the psychosocial correlates of colorectal cancer screening via colonoscopy among the Saudi Population. So, the present study highlighted the psychosocial correlates of CRC screening and its associated factors among the southwestern Saudi population.

A strong and positive sense of coherence is essential for developing a positive attitude towards health, including colorectal cancer prevention and treatment [31]. Two-thirds of the current study participants believed that CRC screening is important to their health. However, a study in Thailand showed that a slightly higher proportion of the participants expressed interest in CRC screening [32]. Also, a study in South Korea confirmed that 80% of respondents accepted CRC screening [33]. The present study revealed that male gender, government sector employees, and high-income participants had significantly positive perceptions and beliefs towards CRC. This indicates that target-oriented health education is necessary for the pre-eligible population with significantly poor perceptions and lesser beliefs about CRC screening. In contrast to the current study results, a cross-sectional study conducted by Nakajima et al. in 2021 did not find a significant correlation with gender and income [24]. The possible factors for this dissimilarity might be the study participants’ incorporations and study settings. The present study included all patients aged 50 years and above, while the latter had participants aged 18 years and above.

Cancer worries are an important psychological factor to be considered while implementing the CRC screening program. A little worry may encourage participants to uptake CRC. This finding has also been observed in breast cancer and prostate cancer screening programs [34,35]. However, moderate-to-high cancer worries, specifically worries related to diagnosis, treatment, and outcomes, may be significant deterrents for CRC screening activities [36,37]. About two-thirds of participants are scared to obtain an abnormal test result. These findings approximate data from a study conducted in England where nearly half of the sample were worried occasionally or sometimes about CRC [38].

The current study revealed that gender and occupation were the most significant factors associated with cancer worries identified via regression analysis. No other socio-demographic or health-related characteristics were significantly associated with the cancer worries construct. Similar to the present study, a nationwide Swedish survey reported a significant association of gender with cancer fears and anxiety towards the CRC screening program. Interestingly, they also found decreased stress and cancer worries towards CRC screening among participants living with partners and working people [39]. Another study recently completed by Kotzur M et al. reported that fear and anxiety about CRC screening test results were significant barriers to the uptake of CRC screening [22]. A study by Quaife S et al. reported that more cancer worries were significantly associated with higher participation in CRC uptake, and there was no association (after adjusting sociodemographic factors) with breast cancer screening participation by women aged 50 to 70 [40]. Our research findings and other studies from different parts of the world indicate that cancer worries, anxiety, and fear of the results of CRC screening are a global phenomenon and require immediate attention that is suitable for local settings to alleviate the major barriers of CRC screening.

Perceived susceptibility refers to an individual’s perception of the development of illness and diseases due to risk factors [41]. The successful implementation of any public health screening program depends upon the public’s awareness of risk factors and their susceptibility to developing diseases [25,42,43]. Some authors evaluated the importance of perceived susceptibility in the breast cancer screening program and revealed that the uptake of breast cancer screening was significantly associated with participants with higher perceived susceptibility [44,45]. Regarding perceived susceptibility, another study found that more than half of participants thought they had a high chance of developing CRC, while only 32% of Western Australians believed the same [46]. The present study results revealed that nearly half of the respondents did not agree with all the statements of the perceived susceptibility construct. In a study in Iran, individuals obtained relatively poor scores for perceived susceptibility [18]. In the current survey, the mean scores were significantly higher among male gender, married participants, those who live in urban areas, and smokers. Similar to the present study, a survey conducted in the Arab region and another study conducted by Maynard, O.M et al. reported that perceived susceptibility to the development of cancer and other tobacco-related diseases was significantly higher among the male gender and smokers [47,48]. Interestingly, a survey conducted by Wongtawee et al. in 2021 in Thailand did not find any significant difference between study groups [49]. Another study conducted in China reported that participants had higher perceived susceptibility towards colonoscopy than other screening methods [50]. The differences between our study and other studies could be the inclusion of study participants, the socio-cultural differences (conservative society), and the survey tools used. Concerning response efficacy, the present study showed that 65.4% strongly agreed or agreed think that if colorectal cancer is diagnosed early, it can be cured, compared to 81.5% in the Australian study.

Social norms and factors play a significant role in the uptake of any screening program implemented by the healthcare organization in a country [51]. Social norms change the motivation and behavior of people depending on the social environment. This can be determined as the perceived prevalence of people’s behavior and the perceptions of how people consider or evaluate a behavior, which may be derived from many sources, such as the people, and the organizations and industries around them [52]. Regarding the social influence construct of the present study, nearly two-thirds of the participants agree that they prefer to follow the advice of their immediate family members, health professionals, and doctors. Similar to the present study, Kelly KM et al. reported that a primary care physician’s support of CRC screening boosts participation rates [53]. In another study, positive perceived family support was related to having positive beliefs about CRC screening [54]. Our findings related to the social influence on the public attitude to the use of colorectal screening are supported by several studies conducted in the KSA and other parts of the world [51,55,56,57]. Interestingly, some studies assessed the social norms and influence of family members on the uptake of breast cancer screening procedures by the eligible population and revealed that low participation was significantly associated with women who received low social and family support to undergo screening procedures [58,59].

## 5. Strengths and Limitations of the Study

The present survey used a validated questionnaire that assessed important public health issues in the KSA. To date, this survey is the one conducted with the highest number of study participants in the KSA. However, certain limitations need to be noted while reading and interpreting the current research findings. Firstly, this cross-sectional survey was conducted in a single region of the KSA. The KSA consisted of thirteen provinces with wide socio-cultural variation among the population in these provinces. Hence, the findings cannot be generalized to the total population of the KSA. Finally, the possibility of bias related to self-reported surveys cannot be ignored.

## 6. Conclusions

In this study, except for the social influence construct, we found that less than half of the participants agreed on most of the five constructs’ statements. The psycho-social correlates of CRC screening are affected by socio-demographic and health-related characteristics of the studied participants. The present study findings also indicate that social influence through immediate family members and health professionals plays a significant role in participants’ intention to receive CRC screening. The current survey results could help policymakers to implement focused health education programs to increase the awareness of CRC risks, the necessity of early diagnosis, and available screening programs in the KSA.

## Figures and Tables

**Table 1 healthcare-11-02791-t001:** Socio-demographic and health-related characteristics of the population (*n* = 790).

Characteristics	Frequency (*n*)	%
Age (mean ± SD)	59.06 ± 6.46	
Gender		
Male	464	58.7
Female	326	41.3
Marital status		
Married	369	46.7
Single	346	43.8
Divorced/widowed	75	9.5
Education level		
No formal education	309	39.1
Up to high school level	330	41.8
University and above	151	19.1
Occupation		
Government	224	28.4
Private	196	24.8
Retired	215	27.2
Unemployed	155	19.6
Monthly family income (in SAR)		
<5000	337	42.7
5000–7000	352	44.6
>7000	101	12.8
Residence		
Urban	503	63.7
Village	287	36.3
Exercise duration per day (in minutes)		
<30	278	35.2
30 to 60	342	43.3
>60	170	21.5
Smoking status		
No	571	72.3
Yes	219	27.7
Presence of chronic disease (such as diabetes, hypertension, and heart disease)		
No	535	67.7
Yes	255	32.3

**Table 2 healthcare-11-02791-t002:** Participants’ responses to the different items of psycho-social correlates (*n* = 790).

Item	Strongly Agree *n* (%)	Agree *n* (%)	Neutral *n* (%)	Disagree *n* (%)	Strongly Disagree *n* (%)
Construct: Salience and Coherence					
Colorectal cancer screening makes sense to me.	303 (38.4)	213 (27.0)	123 (15.6)	80 (10.1)	71 (9.0)
It is important for me to get screened for colorectal cancer.	226 (28.6)	268 (33.9)	143 (18.1)	105 (13.3)	48 (6.1)
Colorectal cancer screening can protect my health.	230 (29.1)	282 (35.7)	166 (21.0)	73 (9.2)	39 (4.9)
If I avoid colorectal cancer screening, I will still be healthy.	201 (25.4)	280 (35.4)	170 (21.5)	95 (12.0)	44 (5.6)
Construct: Cancer Worries					
After screening, I am scared that I will get an abnormal test result.	188 (23.8)	282 (35.7)	178 (22.5)	94 (11.9)	48 (6.1)
I am afraid that screening test result will show that I have colorectal cancer or polyps.	199 (25.2)	290 (36.7)	167 (21.1)	91 (11.5)	43 (5.4)
Construct: Perceived Susceptibility					
The chances of me getting colorectal cancer is high.	189 (23.9)	271 (34.3)	177 (22.4)	105 (13.3)	48 (6.1)
Compared with other persons my age, I am at lower risk for colorectal cancer.	185 (23.4)	287 (36.3)	188 (23.8)	86 (10.9)	44 (5.6)
It is very likely that I will develop colorectal cancer or polyps.	187 (23.7)	290 (36.7)	165 (20.9)	96 (12.2)	52 (6.6)
The chances of me getting colorectal polyps is high.	187 (23.7)	272 (34.4)	181 (22.9)	91 (11.5)	59 (7.5)
Construct: Response Efficacy					
Colorectal cancer can be prevented if polyps are found and removed.	219 (27.7)	285 (36.1)	170 (21.5)	74 (9.4)	42 (5.3)
If colorectal cancer is diagnosed early, it can be cured.	228 (28.9)	288 (36.5)	157 (19.9)	69 (8.7)	48 (6.1)
Construct: Social Influence					
I want to follow my immediate family’s advice about what I should do about screening for colorectal cancer.	217 (27.5)	296 (37.5)	160 (20.3)	74 (9.4)	43 (5.4)
My immediate family thinks I should have screening for colorectal cancer.	216 (27.3)	292 (37.0)	150 (19.0)	92 (11.6)	40 (5.1)
My physician thinks I should have screening for colorectal cancer.	210 (26.6)	290 (36.7)	168 (21.3)	78 (9.9)	44 (5.6)
I want to follow my physician’s advice about what I should do about screening for colorectal cancer.	232 (29.4)	291 (36.8)	150 (19.0)	78 (9.9)	39 (4.9)

**Table 3 healthcare-11-02791-t003:** Relationships of participants’ characteristics with the salience and coherence construct and cancer worries construct (*n* = 790).

	Salience and Coherence Construct		Cancer Worries Construct	
Characteristics	Mean ± SD/r Value	*p* Value	Mean ± SD/r Value	*p* Value
Age	0.631 *	0.012	0.112 *	0.416
Gender				
Male	15.23 ± 4.14	0.001	7.38 ±2.15	0.027
Female	14.16 ± 4.34		7.04 ± 2.08	
Marital status				
Married	15.17 ± 3.99	0.270	7.26 ± 2.15	0.956
Single	14.33 ± 4.41		7.23 ± 2.06	
Divorced/widowed	15.07 ± 4.63		7.19 ± 2.234	
Education level				
No formal education	15.14 ± 4.25	0.054	7.41 ± 2.08	0.090
Up to high school level	14.78 ± 4.26		7.21 ± 2.14	
University and above	14.12 ± 4.21		6.95 ± 2.15	
Occupation				
Government	15.62 ± 3.82	0.030	7.60 ± 2.03	0.021
Private	14.56 ± 4.47		7.17 ± 2.19	
Retired	14.68 ± 4.59		7.01 ± 2.31	
Unemployed	14.05 ± 3.94		7.12 ± 1.86	
Monthly family income (in SAR)				
<5000	14.80 ± 4.20	0.043	7.32 ± 2.10	0.620
5000–7000	14.52 ± 4.41		7.16 ± 2.16	
>7000	15.72 ± 3.78		7.24 ± 2.10	
Residence				
Urban	14.97 ± 4.05	0.124	7.37 ± 2.04	0.203
Village	14.48 ± 4.58		7.01 ± 2.25	
Exercise duration per day (in minutes)				
<30	15.24 ± 4.24	0.080	7.39 ± 2.22	0.302
30 to 60	14.63 ± 4.16		7.18 ± 2.03	
>60	14.39 ± 4.42		7.10 ± 2.15	
Smoking status				
No	14.81 ± 4.17	0.885	7.18 ± 2.15	0.196
Yes	14.76 ± 4.47		7.40 ± 2.06	
Presence of chronic disease (such as diabetes, hypertension, and heart disease)				
No	14.71 ± 4.29	0.428	7.20 ± 2.13	0.432
Yes	14.96 ± 4.17		7.33 ± 2.13	

* r value from Pearson’s correlation test.

**Table 4 healthcare-11-02791-t004:** Relationships of participants’ characteristics with the perceived susceptibility and response efficacy constructs (*n* = 790).

Characteristics	Perceived Susceptibility		Response Efficacy	
	Mean ± SD/r Value	*p*-Value	Mean ± SD/r Value	*p*-Value
Age	0.776 *	0.010	0.191 *	0.208
Gender				
Male	14.63 ± 4.17	0.012	7.66 ± 2.09	0.01
Female	13.88 ± 4.09		7.15 ± 2.17	
Marital status				
Married	14.38 ± 4.10	0.023	7.65 ± 2.07	0.091
Single	14.29 ± 4.09		7.33 ± 2.09	
Divorced/widowed	14.17 ± 4.69		7.01 ± 2.53	
Education level				
No formal education	14.67 ± 4.30	0.036	7.64 ± 2.16	0.162
Up to high school level	14.11 ± 4.06		7.22 ± 2.13	
University and above	14.06 ± 3.99		7.54 ± 2.06	
Occupation				
Government	15.09 ± 3.85		7.86 ± 1.92	0.301
Private	14.39 ± 4.02	0.002	7.24 ± 2.17	
Retired	13.71 ± 4.55		7.16 ± 2.28	
Unemployed	13.96 ± 4.01		7.51 ± 2.09	
Monthly family income (in SAR)				
<5000	14.41 ± 4.19	0.621	7.55 ± 2.09	0.278
5000–7000	14.34 ± 4.18		7.43 ± 2.16	
>7000	13.95 ± 3.92		7.17 ± 2.15	
Residence				
Urban	14.65 ± 4.01	0.003	7.67 ± 2.02	0.001
Village	13.74 ± 4.34		7.01 ± 2.25	
Exercise duration per day (in minutes)				
<30	14.30 ± 4.42	0.935	7.57 ± 2.22	0.336
30 to 60	14.37 ± 3.92		7.44 ± 1.99	
>60	14.24 ± 4.17		7.26 ± 2.26	
Smoking status				
No	14.08 ± 4.12	0.041	7.35 ± 2.18	0.088
Yes	14.95 ± 3.92		7.70 ± 2.14	
Presence of chronic disease (such as diabetes, hypertension, and heart disease)				
No	14.23 ± 4.26	0.358	7.35 ± 2.18	0.039
Yes	14.51 ± 3.90		7.66 ± 2.01	

* r value from Pearson’s correlation test.

**Table 5 healthcare-11-02791-t005:** Relationships of socio-demographic and health-related characteristics with the social influence construct (*n* = 790).

Characteristics	Mean ± SD/r Value	*p*-Value
Age (mean ± SD)	0.564 *	0.021
Gender		
Male	15.34 ± 4.01	<0.001
Female	14.19 ± 4.08	
Marital status		
Married	15.25 ± 3.91	0.043
Single	14.56 ± 4.10	
Divorced/widowed	14.39 ± 4.62	
Education level		
No formal education	15.20 ± 4.14	0.073
Up to high school level	14.48 ± 4.13	
University and above	15.03 ± 3.78	
Occupation		
Government	15.64 ± 3.72	0.008
Private	14.46 ± 4.24	
Retired	14.52 ± 4.44	
Unemployed	14.75 ± 2.70	
Monthly family income (in SAR)		
<5000	15.14 ± 4.04	0.266
5000–7000	14.70 ± 4.12	
>7000	14.56 ± 4.04	
Residence		
Urban	15.25 ± 3.89	0.001
Village	14.20 ± 4.31	
Exercise duration per day (in minutes)		
<30	15.04 ± 4.29	0.671
30 to 60	14.77 ± 3.86	
>60	14.68 ± 4.15	
Smoking status		
No	14.71 ± 4.15	0.079
Yes	15.28 ± 3.85	
Presence of chronic disease (such as diabetes, hypertension, and heart disease)		
No	14.68 ± 4.17	0.053
Yes	15.26 ± 3.87	

* r value from Pearson’s correlation test.

**Table 6 healthcare-11-02791-t006:** Findings of regression analysis assessing relationships of socio-demographic and health-related characteristics with the social influence construct (*n* = 790) **.

Characteristics	Salience and Coherence		Cancer Worries		Perceived Susceptibility		Response Efficacy		Social Influence	
	Regression Coefficient (β)	*p*-Value	Regression Coefficient (β)	*p*-Value	Regression Coefficient (β)	*p*-Value	Regression Coefficient (β)	*p*-Value	Regression Coefficient (β)	*p*-Value
Age	0.151	0.014 *	0.018	0.618	0.025	0.484	0.053	0.131	0.044	0.212
Gender	0.188	0.017 *	−0.150	0.001 *	−0.157	0.022 *	−0.073	0.047 *	−0.104	0.005 *
Marital status	−0.016	0.654	0.013	0.714	0.005	0.893	−0.077	0.062	−0.058	0.107
Education level	0.089	0.061	0.081	0.302	0.073	0.049 *	0.029	0.437	0.031	0.398
Occupation	−0.111	0.002 *	−0.080	0.030 *	0.097	0.008 *	−0.031	0.389	−0.037	0.313
Monthly family income (SAR)	0.065	0.067	−0.005	0.893	−0.010	0.787	0.038	0.282	−0.040	0.261
Residence	0.033	0.354	0.064	0.075	−0.081	0.024 *	−0.130	0.001 *	−0.110	0.002 *
Exercise duration per day (in minutes)	−0.057	0.115	−0.035	0.341	0.015	0.683	−0.038	0.294	−0.009	0.794
Smoking status	0.016	0.645	0.029	0.423	0.076	0.033 *	0.056	0.117	0.045	0.207
Presence of chronic disease (such as diabetes, hypertension, and heart disease)	−0.047	0.194	−0.052	0.152	−0.053	0.144	−0.092	0.011 *	−0.086	0.071

* Significant *p*-value (two tailed). ** Multilinear regression analysis was performed to assess the relationship of each construct with the socio-demographic and health-related characteristics.

## Data Availability

The data used to analyze the present study findings will be provided by the corresponding author on request.

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
