# Peer review of "Assessment of Psychosocial Correlates and Associated Factors of Colorectal Cancer Screening among Southwestern Saudi Population: A Cross-Sectional Study"

_healthcare, 2023, doi:10.3390/healthcare11202791_

Round 1

Reviewer 1 Report

It is stated in the title of psychosocial correlations, but demographic variables and... are also observed in the results.

In the abstract, the purpose of the study should be stated.

Why aged 45 years and above?

Please see STROBE checklist: cross-sectional studies and Ethical consideration should be at the end of the method.

This doi: 10.1016/j.aogh.2016.10.004. Can be useful in introduction (To present the status of colorectal cancer in Asia).

The present cross-sectional study was conducted from April 2022 to November 2022, in this time Covid 19 is an emergency situation, and has disrupted health services. doi: 10.1007/s12029-021-00679-x. The results of this article can be considered in the discussion.

The results of Table 2 are not discussed.

In table 3 what’s Salience and Coherence construct and Cancer Worries construct? Also Perceived suseptibility and Response efficacy?

Why regression analysis was not done?

Conclusions is long.

Reviewer 2 Report

This cross-sectional study explored the influence of psychosocial factors on colorectal cancer screening in a specific population (aged 45 years and above). It aimed at identifying these factors' role in the decision-making process of undergoing CRC screening.

The results have practical implications for policymakers aiming to increase the uptake of CRC screening. However, below are my comments and concerns:

L118: How about if the participant had a "family history" of CRC?

L129: Is this part of the research protocol? "Furthermore, all the data collectors were trained in a standardized way to give health education to the participants after completing the survey forms on the CRC risk factors and prevailing policy on the screening program available in the KSA."

L133: "Firstly, the invited participants were given a brief orientation about different CRC screening methods, the type of screening method recommended by the ministry of health, KSA for high-risk groups, and current survey purposes" Again, is this part of the research protocol? How do you make sure this did not affect their responses? Please mention the setting where the data collection was taking place. How the participants were approached?

L138: "principal author of the survey" is a confusing statement

L241: This is totally irrelevant "World cancer day is held every year on the 4th of February by the Union for International Cancer Council (UICC) to raise awareness of cancer, improve the early diagnosis,  prevention, and treatment for all, and decrease the illness and deaths due to cancer (29).

L311: This is not a limitation.. "Secondly, this research attempted to find the association, not the causation"

Results

The fact that less than 50% agreed with most of the statements may indicate issues with the phrasing or comprehension of the questionnaire items or a strong presence of barriers/misconceptions.

Moderate editing of the English language is required.

Reviewer 3 Report

Prevention mechanisms in any country are important for the early prevention of any type of diagnosis. I suggest the following changes:

#1: In the discussion I would have liked to have related to another type of cancer such as breast cancer. They talk about prevention and I think that breast cancer is one of the most socially invested in social policies in any country and there are many articles written about it.

#2: The conclusion seems to me that it should reflect more what is intended in the article, it seems unclear to me. I would also consider revising the discussion in terms of literary comparison with more articles on the subject.

Round 2

Reviewer 2 Report

No further comments